

# *Buscando Luciérnagas*: findings on Mexican fireflies from an 8-year virtual citizen science project

Ek del-Val, Ana M. Flores-Gutiérrez, Regina González and Adrián Calleros

Instituto de Investigaciones en Ecosistemas y Sustentabilidad, Universidad Nacional Autónoma de México, Morelia, Michoacan, Mexico

Corresponding author
Ek del-Val, ekdelval@iies.unam.mx

## ABSTRACT

Fireflies are charismatic and conspicuous animals that often evoke childhood memories, which make firefly watching an emotional and even transformative experience. Citizen science projects have the potential to enhance transformative interactions with nature. Like many insects, firefly populations are declining due to land-use change, urbanization and watershed pollution, but ecological data for this group is scarce, particularly in Mexico. Virtual Citizen Science (VCS) initiatives can serve as a scientific instrument, yield reliable and relevant scientific data, and may also offer a platform to promote broader educational outcomes. We established a VCS project to document fireflies through a Facebook page named *Buscando Luciernagas* with the following hashtag in every post *#veobrillar* in 2015. After seven years we complied the gathered data and analyzed the results. We had 647 reports in total, with strong fluctuations from year to year that were correlated with the number of posts and publicity we made each year. The largest number of sightings (319) occurred in 2021, coinciding with a change in our reporting format. Most of the reports came from central Mexico (91.5%), but we had reports from eight states and also received some international reports from nine different countries. Fireflies were most frequently seen in habitats characterized as grasslands (35%) or forests (27%), followed by gardens (17%), vacant lots (9%) and parks (5%) but also paved areas and agricultural lands were reported (3% each). Most citizen scientists reported few fireflies, 1–5 individuals (31%) while only 11% reported more than 50 fireflies per sighting. Our study can serve as a preliminary approach to explore more focused research areas in the future. For example, in areas with no sightings, we could reach out to specific local people to corroborate that there are no fireflies in the region, or in areas with high sightings we could promote conservation measures. Notably, we found it intriguing to discover numerous sightings of fireflies in urban areas, which could offer a potential avenue for further research in urban ecology.

## INTRODUCTION

Our wonderful planet is experiencing a significant extinction crisis related to anthropogenic activities (*Dirzo, Ceballos & Ehrlich, 2022*). This ongoing crisis is becoming increasingly evident for many taxa. Insects were initially thought to be less severely affected than other taxa, but recently, studies and investigators have been calling attention to the fact that several families are severely threatened (*Hallmann et al., 2017*; *van Klink et al., 2020*). To assess the magnitude of these impacts, it is important to have local baseline estimates of biodiversity, but these are not available for many regions and/or taxa, particularly in megadiverse countries, such as Mexico.

### Citizen science for monitoring biodiversity

People who volunteer to participate in some part of the process related to science projects are considered to be community or citizen scientists (CSc). They can play an important role in the growth of scientific knowledge given that their participation provides big data observations across time and space (*Gardiner & Roy, 2022*). In entomology-focused programs, citizen scientists have made new discoveries, participated in conservation efforts, and contributed to species monitoring (*Gardiner & Roy, 2022*). These involvements can help to gather the data needed to achieve United Nations Sustainable Development Goals related to biodiversity protection and restoration, sustainable resource use, capacity building and multi stakeholder partnerships, zero hunger and education; bee-focused projects have been particularly illustrative in this sense (*Koffler et al., 2021*).

Virtual Citizen Science (VCS) initiatives can serve as a scientific instrument, yield reliable and relevant scientific data, and may also offer a platform to promote broader educational outcomes (*Wald, Longo & Dobell, 2016*). Recently, the use of social media has facilitated data collection from more volunteers, even in remote places (*Gardiner & Roy, 2022*). One of the main challenges of citizen science projects is to maintain involvement and collaboration over long periods (*Liberatore et al., 2018*; *Oliveira et al., 2021*) making it difficult to train participants for highly standardized data collection. This may pose a tradeoff between making the project accessible *versus* having high quality data (*Bonney et al., 2014*; *Irwin, 2018*). However, the use of social media for citizen science allows regular multi-way communication with the citizen science community, which can aid in retention of volunteers.

According to a review by *Gardiner & Roy (2022)* there are currently many citizen science projects focused on arthropods that have published their experiences. These have mainly been from the USA and UK, where most of the projects are focused on butterflies (Lepidoptera) and bees (Hymenoptera), followed by beetles (Coleoptera). In a recent article *Lewis et al. (2024)* compiled a list of 21 citizen science projects focused on fireflies around the world and recognize several important inputs from them such as refining the firefly checklist for several countries, the monitoring of an invasive species, the monitoring of rare species, among others.

## Charismatic species

Fireflies are charismatic and conspicuous animals that often evoke childhood memories; this makes the experience of firefly watching joyful and even transformative (*Lewis et al., 2021*; *Fakruhayat & Rashid, 2023*). Human-nature experiences, like firefly spotting, can have a strong positive impact on human wellbeing and conservation attitudes (*Fakruhayat & Rashid, 2023*; *Soga & Gaston, 2016*), and citizen science projects can help promote transformative interactions with nature (*Goldin & Suransky, 2023*).

Fireflies are sensitive to changes in their environment and are often used as an indicator species to monitor the health of ecosystems (*Chen et al., 2021*; *Wang, Cao & Wang, 2022*). Fireflies require a specific type of habitat to thrive, and their presence or absence can provide important information about the quality of the environment (*Kazama et al., 2007*). In Mexico, fireflies (Coleoptera: Lampyridae) used to be common and were part of everyday life, even in cities. However, since they are vulnerable to land-use changes and urbanization, there is a decline in their populations and diversity (*Lewis et al., 2020*; *Pérez-Hernández et al., 2023*). Mexican firefly diversity is quite large, 280 species are currently recognized (*Pérez-Hernández, Zaragoza-Caballero & Romo-Galicia, 2022*; *Zaragoza-Caballero et al., 2023*) from a total of 2,200 global firefly diversity (*Lewis et al., 2024*), and new species continue to be discovered (*Gutiérrez-Carranza & Zaragoza-Caballero, 2024*). Most studies from Mexican fireflies concentrate on taxonomical and morphological aspects of the group while ecological and behavioral investigations have been scarce, therefore there is a need to improve our general knowledge about the group in the country.

To evaluate whether this general perception of firefly decline has some support in different parts of Mexico, we launched a citizen science project called "*Buscando luciérnagas*" (Searching for fireflies) to retrieve information about this group of charismatic Coleoptera from the general public. The specific objectives of this initiative were: (1) to understand the general perception of firefly populations in different parts of Mexico, (2) to collect more general information on firefly habitats (*e.g.*, vegetation type), whether it was near a water source, and if it was in the countryside or in a city, and (3) to engage and educate the public. *Buscando Luciernagas* was established with a participatory and adaptive approach to data collection, which made VCS an interesting option to address a larger and more diverse audience in a wider territory. In 2021, Mexico had 88.6 million internet users (75.6% of its population). Of all the internet users, 71.5% used the web to access social media, mainly Facebook, with 96.8% of users reporting having an active account; for 92% of users, their main source of internet connection was through mobile phones (*IFT, 2016*; *INEGI, 2015*). A study done by the Pew Research Center showed that users of mobile connectivity in emerging economies indicated a positive feeling toward social media's societal effects. In Mexico specifically, 67% said the increasing use of the internet had had a good influence on education (*Pew Research Center, 2019*).
## MATERIALS AND METHODS

### Social media as a citizen science platform

For the period in which the project was established (from 2015 to 2022), Facebook was judged to be the best option to approach future Mexican citizen scientists, as it was a tool they already had, knew, and used in a daily manner, with positive social acceptance. In June 2015, a Facebook page called *Buscando Luciernagas* was created, with the following hashtag in every post *#veobrillar* ("I see them shining"). The account shared information about firefly function, morphology, diversity, conservation status, ways to find them, and general identification. With the page running and followers adding up, we encouraged them to go outside at night and search for fireflies near them, which resulted in the first reports. Under an adaptive approach, changing strategies is seen as a strength rather than a methodological inconsistency. Therefore, we once again asked participants to go outside and report the following information on any fireflies they sighted: 1. Where were they seen? 2. How many were there? 3. Were they near water? 4. Share location (in general terms—not GPS coordinates for security reasons).

From 2021 we designed and launched a web application (https://anaeme.shinyapps.io/BuscandoLuciernagas/) with an interactive map and a questionnaire where people were able to record the location of the sighting, date, approximate abundance, vegetation type, presence of water, and light color. Therefore, in 2021 and 2022 we gathered additional data through the new app.

### Outreach, project promotion and data analysis

From 2015 to 2022, radio interviews, scientific dissemination articles, participation in conferences, and other outreach strategies were used to promote awareness of firefly conservation and increase web traffic on the social media page.

To analyze the relationship between the number of reports and habitat characteristics (water presence, habitat type) we performed Chi2 tests. All analysis and figures were performed in R version 4.2.3 (*R Development Core Team, 2017*). To analyze whether the firefly sightings originated from rural or urban areas, we used the World Urban Areas shapefile provided by *Patterson & Kelso (2012)*. We then counted the number of sightings that fell within urban polygons.

### Participant consent and disclosures

People who participated in our citizen science project knew we were gathering data about their firefly sightings to perform the survey. At the Facebook page in the project information section we have an explicit statement saying "The purpose of this project is to monitor fireflies through citizen science. It arises from the Laboratory of Biotic Interactions in Altered Habitats, part of the Institute of Ecosystems and Sustainability Research (IIES), UNAM, Campus Morelia. The data generated by this project will be public, and the identity of the participating individuals will be kept confidential".

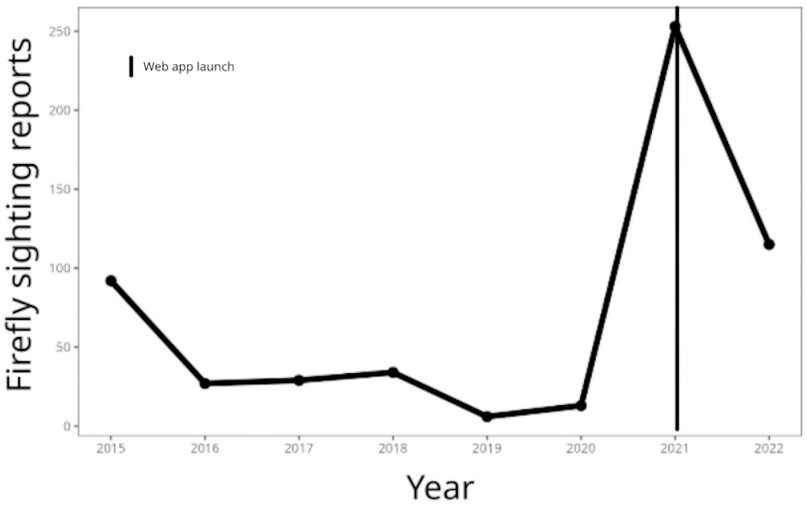

**Figure 1 Temporal trend of firefly sightings reported in *Buscando luciernagas*.** Number of reported sightings from 2015–2022, showing the year when we changed our reporting format.

## RESULTS

We compiled the data from 2015–2022 reporting the sighting of fireflies using social networks through the *#veobrillar*, the *Buscando Luciérnagas* Facebook page, and the web app. We had 647 reports in total (Fig. 1), with strong fluctuations from year to year that were correlated with the number of posts and publicity activities we performed each year. We started with 101 reports in 2015, had the peak in sightings in 2021 with 319 reports, and 116 in 2022.

In total we had 2,707 followers in the Facebook page. Most of the reports came from central Mexico (91.5%), but we had reports distributed in several municipalities from twenty-six states around the country (Fig. 2). We also received a few reports from other countries, such as the USA, Guatemala, El Salvador, Honduras, Canada, Spain, Argentina and Brazil.

Most participants reported only once (18%) or twice (14%). However, we had one person who provided 102 firefly sightings (18%). Additionally, 17 individuals (13%) reported between five and 10 sightings and many participants (38%) were unidentifiable because they did not leave their email contact.

Most firefly sightings were from habitats characterized as grasslands (35%) or forests (27%), followed by gardens (17%), vacant lots (9%) and parks (5%) but also paved areas and agricultural lands were reported (3% each; Fig. 3). Interestingly, according to the World Urban Areas shapefile (*Patterson & Kelso, 2012*) more than half of the firefly sightings in our study came from rural areas (58%) therefore fireflies are significantly more observed outside cities ($X^2 = 16.74$, $p < 0.0001$).

Most participants reported few fireflies, between 1–5 individuals (31%), while only 11% reported more than 50 fireflies per sighting (Fig. 4). The number of fireflies reported per
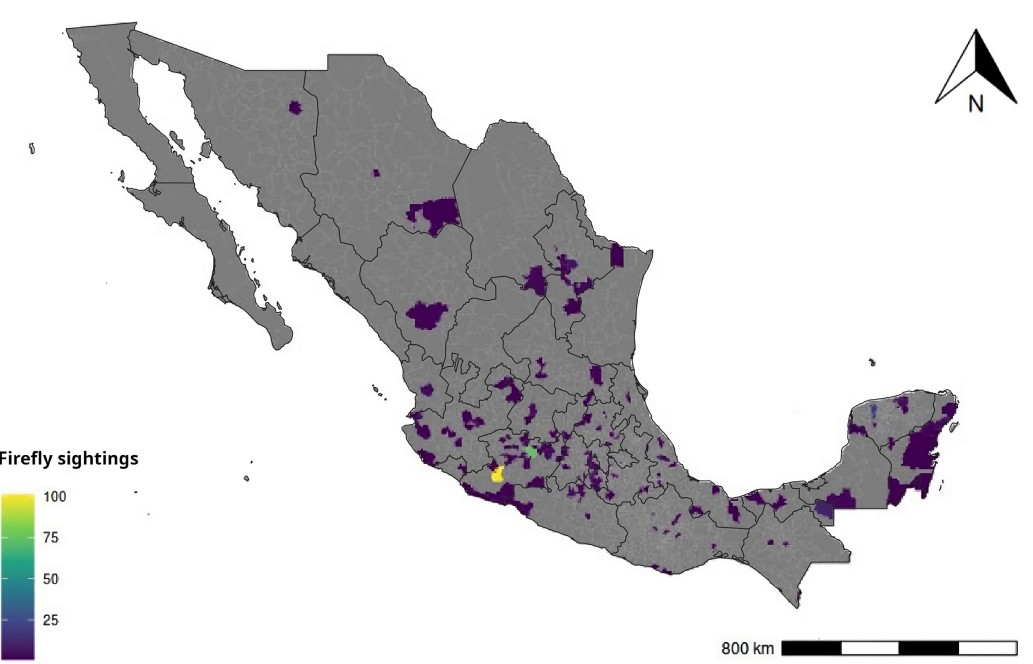

**Figure 2 Locations of firefly sightings throughout Mexico.** Data are shown by municipality. Lighter areas correspond with municipalities with greater number of firefly sightings. Lines in black are State boundaries. Made with Natural Earth. Free vector and raster map data @ naturalearthdata.com.

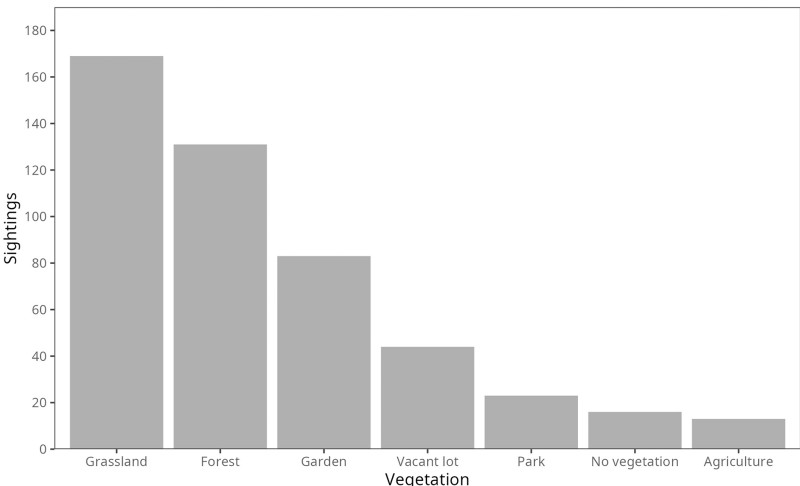

**Figure 3 Firefly sightings per habitat type.** Number of firefly sightings regarding reported habitats.

sighting was significantly different between habitat types; copious fireflies (>50) were mainly found in forests and grasslands, while individual fireflies were reported in all habitat types ($X^2 = 722.6$, $p < 0.001$; Appendix 1A).

When taking into consideration the presence of water sources, interestingly we did not find a correlation between the presence of water and fireflies; half of the reports mentioned

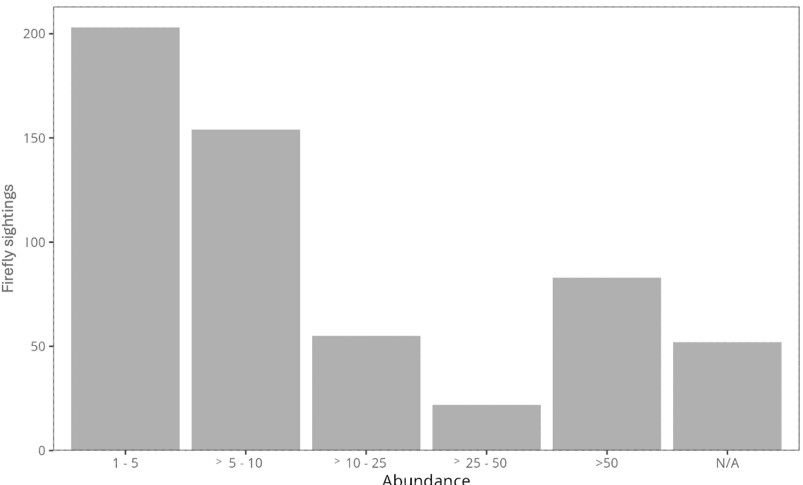

**Figure 4 Firefly sightings per abundance category.** Number of firefly sightings regarding abudance category. N/A corresponds to reports that did not include abundance.

a water source while the rest did not find any. However, there were significantly more reports of copious fireflies in sites with water ($X^2$ = 161.7, $p$ = 0.0005; Appendix 1B).

Firefly light color was a question added to our survey in 2021. However, since most people did not answer this question, we did not analyze that data. We believe that for some people, it is difficult to ascertain the color of the flashing. Our team has discussed this issue, particularly noting that when observing just one or a few fireflies, the color may not be that obvious.

## DISCUSSION

Our *Buscando Luciérnagas* initiative was successful in involving people from around Mexico. We were able to capture the attention of citizen scientists from different states and regions, and we were able to witness their enthusiasm and desire to help firefly conservation and to learn more about their ecology and life history.

Most of our firefly reports coincide with The Mexican Transition Zone, which has been recognized to have the highest Lampyridae species richness and endemism in the country (*Pérez-Hernández, Zaragoza-Caballero & Romo-Galicia, 2022*). However, this pattern also coincides with the zone of Mexico where we (the investigators) are based and where we publicize our initiative in local media, particularly the city of Morelia, Michoacán, and surrounding areas where we compiled 40% of the sightings. Therefore, we cannot rule out the possibility that the pattern found is correlated with our capacity to reach people more than with actual firefly abundance or diversity. On the other hand, it is worth noting the lack of sightings in the Baja peninsula and the coastal northwest, since there are many conserved areas in those places, we thought there would be at least some reports, however this could be related with a low abundance/diversity of fireflies or a lack of citizen scientist involvement in that region. Both questions warrant further investigation.

Since many firefly species are dependent on water sources for their larval development (*Riley, Rosa & da Silveira, 2021*), we expected to find a correlation between firefly sightings

and water sources, yet this was not the general trend: roughly half of the observations came from sites without water sources nearby. When looking into the details of reported abundance however, it is worth noting that the sightings of high firefly abundance (>50 individuals) indeed come from sites near water sources. Therefore, it is very plausible that individual fireflies can be found in most of the studied sites, but healthy populations need some water sources in the vicinity, as has been previously reported for many studied fireflies (*Lewis et al., 2020*). Also, it is possible that some of the sightings came from firefly species that do not rely on running water to complete their life-cycle, however this hypothesis requires further investigation. Unfortunately, we do not have detailed information about the ecology of Mexican firefly species therefore we are not certain about specific reliance on water sources but in the recent study about the firefly fauna in Morelia, *Pérez-Hernández et al. (2023)* reported at least eight species (out of 24) were preferentially associated with water sources (*i.e.*, wetlands, rivers, ponds).

Regarding the habitat type where fireflies were reported, most of them (58%) came from rural areas while 62% came from vegetated habitats including the ones inside cities (*i.e.*, grasslands, forests, parks and gardens; Fig. 3), and only 3% from agricultural lands. A recent study about habitat preferences of the Mexican firefly *Photinus palaciosi* found that firefly abundance was not related with forest type, they were equally thriving in oak, pine and mix forests (*Ramírez-Manzano et al., 2023*) but they did not investigate agricultural sites. It is interesting the trend we found towards having more fireflies in cities than agricultural lands, since we tend to think of agriculture as friendly for biodiversity while cities being completely unfriendly. However, since modern agriculture uses large quantities of pesticides, agricultural plots can be more harmful for biodiversity than cities; this tendency has been found for birds (*MacGregor-Fors, 2008*), frogs (*Valdez, Gould & Garnham, 2021*) and bees (*Theodorou et al., 2020*). Specifically for fireflies, insecticide use and the increase of the agricultural frontier are recognized as important causes for their decline (*Lewis et al., 2020*). However, agriculture is not homogeneous in Mexico. There are traditional cropping systems that use fewer external outputs and can function as important biodiversity reservoirs (*Fonteyne et al., 2023*; *del-Val, Aster & Ramírez, 2021*). In this regard, it is worth mentioning that some ecotourism sites for firefly sightings, such as the Firefly Sanctuary in Tlalpujahua, are located on the margins of traditional maize crops; hence, in order to have substantial conclusions about the effects of firefly abundance in agricultural areas in Mexico, we need more detailed information about the sites.

The reporting peak in 2021 coincided with a change in our reporting format; apparently, the new format was preferred by citizen scientists. Additionally, this period also overlapped with the COVID-19 pandemic, during which anthropogenic activity was considerably reduced, and several animals reportedly increased their activity (*Burton et al., 2024*). Therefore, fireflies could have also benefited from the decrease in human outdoor activities. However, studies investigating the anthropogenic impacts on fireflies have found that the most significant factors affecting their populations are light pollution, land-use change, climate change, and agricultural intensification (*Lewis et al., 2020*; *Fallon et al., 2021*) therefore it is not clear how a reduction on anthropogenic activity could have benefited them.

## Citizen science projects in the advancement of knowledge

While with this initiative we were able to identify the presence of fireflies at a Family level, we cannot differentiate between species, therefore there is a need for detailed ecological investigations that help to elucidate Mexican firefly species distribution and life history (*Bonney et al., 2009*). In order to do so through CS projects, we will need to generate detailed information about their flashing patterns and communicate it to the public, as has been done in other regions (*Lloyd, 1966*; *Faust, 2017*). Another possibility to gather more specific and reliable data on Mexican fireflies would be to combine virtual citizen science projects with on-field citizen science initiatives. For example, we could select some localities with reported high incidence or where fireflies are commonly observed from our database and train a team of community science volunteers to perform field surveys throughout the year. This strategy has been successful in Hong Kong (*Yiu, 2023*), England (*Gardiner & Didham, 2020*), and Japan (*Takeda et al., 2007*) where citizen scientists gather at specific dates and follow sampling protocols established by academics that can cover large areas and provide more specific and detailed information.

There are two opposing challenges related to Citizen Science. On the one hand, there is a call to make data collection procedures more rigorous to enhance credibility. On the other hand, there is a need to make them accessible and easy for the public to participate. In that sense, although our study was unable to discern between firefly species due to the lack of easily distinguishable characteristics accessible to the general public, the bioluminescent sparks serve to characterize the presence of a firefly from the Lampyridae family, enabling us to generate maps with presence data at this broad taxonomic level. One concern is that the CSc reports may include bioluminescent beetles from the Elateridae family that occur in the study area. Although we cannot rule out the possibility that some reports correspond to this group, we provided information on our Facebook page to help distinguish Elateridae from Lampyridae, and responded to some questions related that were asked on the page. Also, in the radio interviews we talked about the differences between fireflies and other bioluminescent animals. Since fireflies are generally more abundant than bioluminescent click beetles, we are confident that most of our reports are indeed Lampyridae.

Throughout the course of our study, we were able to witness the involvement and excitement of many citizen scientists that participated in our survey. We found an array of experiences, ranging from people that had never seen them in the cities, to people that realized they were not as common as when they were kids. Also, people wanted to know more about bioluminescence and to be involved with their conservation. It was evident to us that people who got involved in this type of project strengthened their link with nature, as other authors have found (*Williams et al., 2021*). With our data, we were able to identify a gradient of involvement among citizen scientists, since a few individuals provided many of our sightings (31%), while most were casual participants, reporting only once or twice during the study period. Therefore, most involvement with our project was casual rather than long-lasting. To improve the relationship between citizen science projects and the general public, we suggest incorporating periodic activities where people can feel that they

are part of a community and that their involvement contributes to scientific advancement. This paper will be shared on the *Buscando luciernagas* Facebook page to disseminate the results and emphasize this community effort.

One important aspect of CSc is that once people get more concerned with nature by participating in these projects, they increase their entomological knowledge and modify their everyday practices, such as gardening, be environmentally friendly, reducing pesticide application and the frequency of mowing (*Fontaine, Fontaine & Prévot, 2021*). We were also able to witness this, as many participants who participated returned the following year to upload their new sightings. Also, people continuously ask about environmental measures to take in their communities to improve conditions for fireflies. For example, a woman requested "we want to know which plants we can sow in our environment to attract them, last year there were many but not this one." Another citizen asked "what can I do for their preservation?"

An interesting finding was that several people that were interested in our social media campaign reached out to us to share that they used to see fireflies in their communities/ environments in the past but were not able to see them any longer. One citizen, for example, said "it has been a long time since I have seen any" and they were interested to know if there were specific conservation/ sanctuary areas where they could go. This was a repeated request, highlighting the fact that people are interested in nature contemplation.

Our approach to gathering Mexican firefly data through a virtual citizen science project not only helped us gain a general understanding of the distribution of these coleopterans in certain regions but was also particularly successful in engaging the public to observe the insects and reflect on their current conservation status. This survey can be helpful to define areas to further sampling as mentioned above and to explore in more detail the correlations we found. We recommend that future virtual citizen science projects aiming to get more specific data should include some form of virtual training, such as webinars or online tutorials, to increase the level of confidence in the data collected. Since a couple of years, there had been more events related with the International Firefly Day in the city of Morelia, where the project started, and people are engaging with them at the point in which entrances are usually sold out, which demonstrates that this kind of citizen science projects can lead people to engage with nature.

## CONCLUSIONS

Our findings support previous assertions regarding the significance and benefits of citizen science: it can simultaneously cover various locations, span large study areas, and have local impacts. Our study lays the groundwork for further investigation into more specific research areas in the future. For instance, in regions devoid of firefly sightings, we could engage with local communities to confirm the absence of fireflies. Conversely, in areas with high sighting rates, we could advocate for conservation measures. Particularly intriguing was the abundance of firefly sightings in urban settings, suggesting a promising avenue for urban ecology research. This phenomenon underscores the potential of cities as biodiversity reservoirs and emphasizes the importance of designing urban spaces to be conducive to biodiversity conservation.

## ACKNOWLEDGEMENTS

We are immensely grateful for all the marvelous citizen scientists who participated in our study; without them, this would have been impossible. Additionally, we want to extend our gratitude to all the local media outlets that helped us reach a broader audience. We thank Elizabeth Flores Jurado for the drawings of the fireflies. Finally, we thank the translator Lynna Kiere for revising our manuscript. We appreciate the valuable comments provided by three anonymous reviewers.

### Funding

The authors received no funding for this work.

### Competing Interests

The authors declare that they have no competing interests.

### Author Contributions

- Ek del-Val conceived and designed the experiments, performed the experiments, analyzed the data, authored or reviewed drafts of the article, and approved the final draft.
- Ana M. Flores-Gutiérrez conceived and designed the experiments, performed the experiments, analyzed the data, authored or reviewed drafts of the article, and approved the final draft.
- Regina González conceived and designed the experiments, performed the experiments, analyzed the data, authored or reviewed drafts of the article, and approved the final draft.
- Adrián Calleros analyzed the data, prepared figures and/or tables, authored or reviewed drafts of the article, and approved the final draft.

### Data Availability

The raw data gathered through Facebook is available in the Supplemental File.

### Supplemental Information

Supplemental information for this article can be found online at http://dx.doi.org/10.7717/peerj.18141#supplemental-information.

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
