# Peer review of "Buscando Luciérnagas: findings on Mexican fireflies from an 8-year virtual citizen science project"

_PeerJ, doi:10.7717/peerj.18141_

## Round 0.1 · original submission · Major Revisions

Dear Dr. del-Val and colleagues:

Thanks for submitting your manuscript to PeerJ. I have now received three independent reviews of your work, and as you will see, one reviewer recommended rejection, while the others suggested revisions (with many suggested changes). I am affording you the option of revising your manuscript according to all reviews but understand that your resubmission may be sent to at least one new reviewer for a fresh assessment (unless the reviewer recommending rejection is willing to re-review).

The reviewers raised many concerns about the manuscript. Please address all of these in your rebuttal letter. I especially would like to see your response to concerns over a lacking clear hypothesis/research question, as well as an uncertain taxonomic scope for a “biodiversity” study.

There are many minor suggestions to improve the manuscript. Note that reviewer 3 kindly provided a marked-up version of your manuscript.

Therefore, I am recommending that you revise your manuscript, accordingly, considering all of the issues raised by the reviewers.

Good luck with your revision,

-joe

Reviewer 1 ·

Basic reporting

This manuscript describing a successful citizen science initiative in Mexico, Buscando Luciernagas, is clearly written and well-structured, and the authors have done a good job of setting the context for this work. Appropriate references are provided.

Experimental design

This observational study fills a knowledge gap about citizen science reporting of fireflies in Mexico, and methods for collecting data are nicely described. The specific research question and hypothesis for this study could be more clearly defined: was it to evaluate a “general perception of firefly decline in different parts of Mexico (lines 103-4)? To collect more general information on firefly habitats & distribution (lines 106-108)? To engage and educate the public?

Validity of the findings

3a. It is not clear whether observers were instructed or trained in distinguishing between firefly beetles (i.e. Lampyridae) & other bioluminescent beetles (particularly Elateridae). According to iNaturalist, there are several genera of click beetles (Deilelater, Pyrophorus, Ignelater) that commonly occur in this geographic area. Perhaps it could be specified if these were included in citizen reports.

3b. In the Discussion it would be worthwhile to explicitly mention the possibility of observer bias – for example 34% of reports were from cities, presumably because the majority of respondents live in urban areas.

Additional comments

Below are few suggestions that I hope might be useful in revising the manuscript.

Major comments:
1. This appears to be the first firefly citizen science project conducted in Mexico, and it was clearly highly successful in engaging many people. However, it is worthwhile pointing out that numerous firefly citizen science projects exist in other regions; in a 2024 review, a compilation of ~20 firefly citizen science projects is given in Supplementary Table S2 (https://www.mdpi.com/article/10.3390/insects15010071/s1) (https://doi.org/10.3390/insects15010071). It is understandable that this is not referenced in the current article, as perhaps it was not yet published. Also, it is worth noting that neither of the two US firefly projects mentioned on lines 80-85 remains active.

2. Figure 1 (reports by year) could be made more informative by annotating the year axis to include the methodological changes mentioned in the Materials & Methods.

Minor comments:
Line 121 “proved” might be replaced by “was judged”
Line 197 tiny typo: compiled

·

Basic reporting

Overall, the writing in English is generally clear and grammatically correct. However, there are areas where clarity could be improved by ensuring that terms translated from Spanish make sense in English. For example, it is not clear to me what “again” means in the context of the title (line 1), and the authors use the word “publication” instead of “posts” when discussing social media content (lines 24, 125, 162).
Literature references, sufficient field background/context provided.

The authors give provide an adequate literature review to provide context about insect declines and the role of citizen science in monitoring biodiversity. However, I think the authors could and should have delved deeper into the literature on firefly citizen science. The introduction lists two firefly citizen science projects, but there are many more (see supplementary materials of Lewis et al. 2024, https://www.mdpi.com/2075-4450/15/1/71 for a table of over twenty firefly-related citizen science projects). While a full review of these projects is their resulting publications is beyond the scope of this article, there is a rich opportunity for framing “Buscando luciérnagas” in terms of opportunities, challenges, and best practices in firefly citizen science.

More context about the state of knowledge regarding firefly diversity and conservation status in Mexico would be helpful. For example, how approximately how many species are described from the country? Are any fireflies legally protected? Are any species or populations being studied more closely?

While the article follows standard scientific article structure, I think that it would benefit from subheadings to guide the reader. For example, under Materials & Methods, subheadings could include “Social media as a citizen science platform,” “Questionnaire Design and Implementation,” “Outreach and project promotion,” “Data analysis and visualization,” and “Participant consent and disclosures.”
The figures were all intelligible, but their captions and axis labels could provide more detail such that the figures can stand alone. For example, the y-axis of Figure 1 could be labeled “Firefly sighting reports” rather than simply “sightings,” and the figure caption could specify that firefly sightings were reported through the Facebook page and the shiny-app questionnaire. Including the year that the shiny app was launched in the caption would help the reader interpret the large increase in submissions from 2020 to 2021.

Similarly, the map in Figure 2 would benefit from a north arrow and a scale bar. The significance of the municipalities that are labeled with call-out boxes is not readily apparent. Given the stated importance of outreach for promoting the project, it might be helpful to label the locations of in-person outreach events and to mention them in the figure caption.

This paper did not ask a specific question and then test a hypothesis. Rather, it reports on a data-set compiled through citizen science that could be used and analyzed in various ways. While the article does provide some patterns regarding the habitats and abundance categories of firefly sighting reports, the lessons of the study are just as much about the process of engaging the public and gathering firefly data across a broad geographic area with the help of non-scientists.

I think that the data-set could have been explored, analyzed and visualized more thoroughly.
For example, how many unique contributors to the project were there? Did most participants contribute a single report, or did most contributors submit multiple reports? A figure of firefly sightings by week of year or month would also be helpful and interesting. I don’t think the article needs to explore the data-set from every possible angle, but the article could further investigate patterns in the data. For example, do null values in the data set point to types of data that are difficult for average citizen science participants to collect (eg. abundance)?

While the gender ratio of Facebook followers was intriguing, it did not seem appropriate to report in the results because collecting or analyzing this data was not mentioned in the Methods, and it is unclear whether the Facebook followers consented to having their personal information (gender identify) analyzed.

Experimental design

The article presents original work about a data set gathered through a citizen science project that was implemented by the authors.

As stated previously, this article does not explicitly ask a research question or set of questions. Rather it reports on an exploratory gathering of firefly citizen science data in Mexico, which evolved from informal reports on social media to a more structured online questionnaire. While it is reasonable for a citizen science project to have “imperfect” experimental design and for reporting forms to evolve and adapt organically, I would like to see more discussion about appropriate uses for this data-set and its limitations, as well as recommendations for future firefly citizen science initiatives.

The authors do clearly explain why firefly citizen science is relevant and meaningful, both for gathering data and increasing interest and motivation for conservation of fireflies and other insects.

For the most part, the authors make it clear that participation in Buscando Luciérnagas involved informed consent and that the privacy and safety of participants was a priority.

Validity of the findings

The authors present summaries and patterns regarding a firefly sighting report data set compiled through citizen science in Mexico. The authors were appropriately cautious in interpreting their findings regarding distribution, habitat associations and abundance. A discussion of potential sample biases of project participants would further put the results in context.

Additional comments

I think that the title could be much more specific and compelling.
Something like "Buscando Luciernagas: findings on Mexican fireflies from an 8-year virtual citizen science project" would tell the reader much more about the content of the paper.

Reviewer 3 ·

Basic reporting

Clear and unambiguous, professional English used throughout. The article lacks documentation on the importance of bodies of water for different species of fireflies, as well as mentioning which Mexican species would require habitats with bodies of water to complete their life cycle or to carry out their activities. On the other hand, the structure of the article conform to, in general, an accptable format of standard sectios. However, the authors mix up elements of the Discussion a bit in the Results section and do not follow the editorial rules for presenting References. The authors observe a peak of activity in 2021 and attribute it to "the number of publications and publicity activities we performed each year”. They conclude that this peak “coincided with a change in our reporting format”. It is requested to present only the findings in the Results section and discuss them only in the Discussion section. On the other hand, the authors must be critical of this assertion and delve into the reasons for this observed peak. It was not unusual that during the Covid-19 pandemic and low human mobility, the activity of wild animals enjoyed a period of rest from the profound impact that the human species causes on the activity of wild biota (see references in comments document). The authors are obliged to take into account this and other possible factors of the existence of the increase in activity in 2021-2022.

Experimental design

Research question well defined, relevant and meaningful. It was performed to a high technical and ethical standard. On the other hand, Methods described with sufficient detail and information to replicate. A small detail is that the applied questionnaire gives answers of abundance categories that overlap each other (1-5, 5-10, 10-25, etc.): if an observer sees 5 specimens, she/he can report 1-5 or 5-10.

Validity of the findings

All underlying data have been provided and they are robust. Some recommendations were made to improve the statistical analyzes and to provide the reader with greater clarity on how chi2 tests were applied. The authors must exhaust the hypotheses to explain the 2021-2022 peak of activity, which includes the possibility that low human mobility due to the Covid-19 pandemic has allowed the recovery of the populations of these insects.

Additional comments

It is a very interesting and robust study (with 8 years of data), based on the qualities of Citizen Science, which allows us to understand, with certain limitations, as the authors themselves comment, the distribution and abundance of Lampyridae in different habitats. In my opinion, the authors must respond to the criticisms made and resolve the doubts raised for the article to be accepted.

Annotated reviews are not available for download in order to protect the identity of reviewers who chose to remain anonymous.

---

## Round 0.2 · Minor Revisions

Dear Dr. del-Val and colleagues:

Thanks for revising your manuscript. The reviewers are mostly satisfied with your revision (as am I). Great! However, there are some additional concerns still raised by Reviewer 5, and some edits to make. Please address these ASAP so we may move towards acceptance of your work.

-joe

Reviewer 4 ·

Basic reporting

I have revised both the manuscript and the rebuttal letter. My intention was to see whether authors followed those reasonable suggestions so that the new version looks improved. My general impression is that authors did a great job in crafting their work both by accepting those criticisms that made sense as well as adding or removing information they realized was or was not needed.

Given the above, my opinion is that this paper looks great now and ready to be accepted.

Experimental design

This is report of data collected by citizens who watched fireflies (mainly in Mexico) and sent such information via Facebook. This approach is valid.

Validity of the findings

Validity is 100% as authors filtered and analyzed the sightings good enough to sustain their conclusions. I particularly like the data where fireflies are linked to land use as this is very novel. This should serve as an initiative to promote firefly conservation.

Reviewer 5 ·

Basic reporting

The manuscript entittled “Buscando Luciérnagas: Findings on Mexican Fireflies from an 8-Year Virtual Citizen Science Project” describes citizen science project in Mexico. Manuscript is well written and some of the corrections has been made nicely, however there are still some issues that need to be solved to be available for publication in my opinion

My main concern is as previous reviewers that the training for the observers is still to vague, also the objective: to understand the general perception of firefly decline in different parts of Mexico, how was this made?
How methodology was changed from year to year? could you clarify?
Some literature that can help is still missing:
Emerging technologies in citizen science and potential for insect monitoring
Julie Koch Sheard, Tim Adriaens, Diana E. Bowler, Andrea Büermann, Corey T. Callaghan, Elodie C. M. Camprasse, Shawan Chowdhury, Thore Engel, Elizabeth A. Finch, Julia von Gönner, Pen-Yuan Hsing, Peter Mikula, Rui Ying Rachel Oh, Birte Peters, Shyam S. Phartyal, Michael J. O. Pocock, Jana Wäldchen and Aletta Bonn See fewer authors
Published:06 May 2024https://doi.org/10.1098/rstb.2023.0106
A protocol for harvesting biodiversity data from Facebook
Shawan Chowdhury, Sultan Ahmed, Shofiul Alam, Corey T. Callaghan, Priyanka Das, Moreno Di Marco, Enrico Di Minin, Ivan Jarić, Mahzabin Muzahid Labi, Md. Rokonuzzaman, Uri Roll, Valerio Sbragaglia, Asma Siddika, Aletta Bonn … See fewer authors
Also some new literature on Ecology of Lampyrids in Mexico see Influencia de los factores abióticos y del tipo de vegetación sobre la abundancia de los adultos de Photinus palaciosi (Coleoptera: Lampyridae) en Nanacamilpa, Tlaxcala, México DOI:
10.22201/ib.20078706e.2023.94.5091

On the survey you display a question about light colour but it wasnt taken on the results, could you add what happened with this variable?
You are saying that your results are from 8 states, however I see on the map that you have records from at least 15 states of Mexico, my question is that if you only took eight stats? how did you got rid of the other record from the other states?
Can you add to the map the states limits?
I am wondering how the data from the person that reported 179 can overestimate your data? any clue?
Do the yellow part on the map is the place in where the person with the highest number of reports came from?
I was expecting also an statistical analysis contrasting city or coutryside due at the abstract you are talking about urban ecology
I was wondering that an interesting comparison that you can potentially use is to overlap a GBIFF map (Naturalist) with the map you generated to corroborate some of the information that you collect. Naturalist has the advantage that it has photographs from all the insect species.
About the increase on COVID period, I agree that anthropogenic activiry was reducedbut also I would add that most of the people change their habits and they become closer to nature specially on the countryside may be this also was a factor that make people more aware of this marvelous insects
I am still worried about how citizen scientists where trained about the insect identification, I am aware you tried to bring this issue on the discussion but still i suggest that could be a good idea the use of a naturalist map that can help to corroborate/discriminate the presence of the correct insect family

Experimental design

no comment

Validity of the findings

no comment

---

## Round 0.3 · accepted · Accept

Dear Dr. del-Val and colleagues:

Thanks for revising your manuscript based on the concerns raised by the reviewer. I now believe that your manuscript is suitable for publication. Congratulations! I look forward to seeing this work in print, and I anticipate it being an important resource for groups studying fireflies and the citizen science initiatives. Thanks again for choosing PeerJ to publish such important work.

Best,

-joe